# Decidual Stromal Cell Ferroptosis Associated with Abnormal Iron Metabolism Is Implicated in the Pathogenesis of Recurrent Pregnancy Loss

**DOI:** 10.3390/ijms24097836

**Published:** 2023-04-25

**Authors:** Fengrun Sun, Liyuan Cui, Jinfeng Qian, Mengdie Li, Lanting Chen, Chunqin Chen, Dajin Li, Songcun Wang, Meirong Du

**Affiliations:** Laboratory for Reproductive Immunology, Key Laboratory of Reproduction Regulation (Shanghai Institute of Planned Parenthood Research), Shanghai Key Laboratory of Female Reproductive Endocrine Related Diseases, Hospital of Obstetrics and Gynecology, Fudan University Shanghai Medical College, Shanghai 200011, China

**Keywords:** decidual stromal cells, recurrent pregnancy loss, iron, ferroptosis

## Abstract

Iron is necessary for various critical biological processes, but iron overload is also dangerous since labile iron is redox-active and toxic. We found that low serum iron and decidual local iron deposition existed simultaneously in recurrent pregnancy loss (RPL) patients. Mice fed with a low-iron diet (LID) also showed iron deposition in the decidua and adverse pregnancy outcomes. Decreased ferroportin (cellular iron exporter) expression that inhibited the iron export from decidual stromal cells (DSCs) might be the reason for local iron deposition in DSCs from low-serum-iron RPL patients and LID-fed mice. Iron supplementation reduced iron deposition in the decidua of spontaneous abortion models and improved pregnancy outcomes. Local iron overload caused ferroptosis of DSCs by downregulating glutathione (GSH) and glutathione peroxidase 4 levels. Both GSH and cystine (for the synthesis of GSH) supplementation reduced iron-induced lipid reactive oxygen species (ROS) and cell death in DSCs. Ferroptosis inhibitor, cysteine, and GSH supplementation all effectively attenuated DSC ferroptosis and reversed embryo loss in the spontaneous abortion model and LPS-induced abortion model, making ferroptosis mitigation a potential therapeutic target for RPL patients. Further study that improves our understanding of low-serum-iron-induced DSC ferroptosis is needed to inform further clinical evaluations of the safety and efficacy of iron supplementation in women during pregnancy.

## 1. Introduction

Miscarriage is the most common complication of pregnancy. Recurrent pregnancy loss (RPL), also known as recurrent spontaneous abortion, defined as two or more consecutive pregnancy losses, affects up to 5% of women trying to conceive. RPL may seriously compromise not only the physical but also the psychological well-being of women. Although genetic abnormalities, infection factors, endocrine disorders, uterine anatomic abnormalities, and antiphospholipid syndrome are known to be common causes of RPL, the etiology of approximately half of the cases is still unexplained [1]. The underlying cytological and molecular mechanisms for RPL remain largely enigmatic [1].

Human pregnancy is a complex process involving interactions between fetal- and maternal-derived components, as revealed by single-cell sequencing [2]. There is an increasing interest in the role of the decidua in orchestrating the homeostatic balance between the mother and fetus, though the role of the decidua in pregnancy maintenance has largely been neglected. Once conception occurs, the developing decidua undergoes dramatic changes in response to invading trophoblasts in such a way that supports further development or, alternatively, promotes active rejection of the embryo [3]. This quality control, which begins immediately post-conception, probably continues throughout the first trimester of pregnancy. In addition, sonographic images have revealed some differences in first-trimester decidua thickness in miscarried pregnancies compared to normal pregnancies, suggesting that defective development of the decidua could cause miscarriages [4].

Decidual stromal cells (DSCs) are the predominant cell type in the maternal decidua during early pregnancy and play key roles in embryo implantation and placentation [5,6]. Apart from nutritive and endocrine functions, DSCs are believed to be involved in many immune activities, such as cytokine production, antigen presentation, and regulation of the decidual immune responses that may lead to either a successful pregnancy or miscarriage [7]. Excessive senescence or apoptosis of DSCs is associated with RPL [8]. Obstructed communication between DSCs and other cell types has also been observed in RPL patients [6].

Iron is necessary for various critical biological processes involving oxygen transportation, ATP generation (as cofactors of many proteins involved in the tricarboxylic acid cycle and electron transport chain), and DNA biosynthesis (as a ribonucleotide reductase cofactor). Iron deficiency is correlated with several adverse pregnancy outcomes, such as increased maternal illnesses, prematurity, and intrauterine growth restriction [9]. Prenatal iron deficiency is also associated with a number of persistent short-term and long-term developmental deficits of multiple organs, even after iron supplementation. Animal studies have demonstrated that abnormal gene expression and epigenetic changes are related to prenatal iron deficiency in some brain diseases [10]. Iron overload is also dangerous since labile iron is redox-active and toxic. It can generate reactive oxygen species (ROS), which results in oxidative stress and the activation of programmed cell death pathways, such as ferroptosis and iron-related autophagy [11]. Ferroptosis in placental tissues is involved in the pathogenesis of preeclampsia [12], spontaneous preterm birth [13], and gestational diabetes mellitus [14].

The most well-studied process of iron trafficking during pregnancy is through the placenta. Little is known regarding the iron status of the decidua in the context of RPL. Whether iron metabolism regulates DSC function in patients with RPL has also not yet been established. This study aims to investigate the relationship between iron status and DSC function and its effect on pregnancy outcomes by using primary DSCs and different pregnant mouse models, providing a theoretical foundation for improving pregnancy outcomes for RPL patients.

## 2. Results

### 2.1. Iron Deposition in DSCs of RPL Patients and Abortion Mice

To differentiate the iron status between normal pregnancy and RPL patients, we first compared their serum iron content. Serum iron in RPL patients was much lower than that in normal early pregnancies (Figure 1A), although maternal hemoglobin did not differ between the two groups (Figure 1B). Then we tested the intracellular iron level of DSCs by flow cytometry with calcein, whose fluorescence intensity negatively correlates with liable intracellular ferrous iron levels. Interestingly, the percent of calcein^+^ DSCs was lower in RPL patients than in normal early pregnancies (Figure 1C). This result was further confirmed by increased ferritin expression, a surrogate marker for the intracellular iron level, in DSCs from RPL patients (Figure 1D). In addition, transferrin receptor 1 (TfR1), which induces cellular uptake of iron from transferrin (an iron-binding serum protein) by endocytosis, was higher in the RPL decidua, whereas ferroportin (Fpn), the sole membrane iron exporter, was decreased in the RPL decidua (Figure 1E). The increased expression of TfR1 protein and decreased expression of Fpn protein were also shown in DSCs from RPL patients compared to those from normal pregnancies (Figure 1F), suggesting that increased iron deposition in the decidua is associated with RPL.

We established an abortion-prone model using female CBA/J × male DBA/2 mice and found decreased calcein but increased ferritin expression in DSCs of abortion-prone mice (Figure 1G), accompanied by increased *FTH* (Ferritin heavy chain coding gene) and decreased *SLC40A1* (Fpn coding gene) expression in the uterus of abortion-prone mice (Figure 1H). *TFRC* (TfR1 coding gene) expression did not differ between the two groups (Figure 1H). A similar phenomenon was also observed in the lipopolysaccharide (LPS)-induced abortion model (Figure 1I,J). These data imply a correlation between iron deposition in DSCs and an adverse pregnancy outcome, though the serum iron level of RPL patients is low.

### 2.2. Low-Iron-Diet-Induced Iron Deposition in Decidua and Adverse Pregnancy Outcome in Mice

It was striking that low serum iron and decidual local iron deposition existed simultaneously in RPL patients. We fed female C57BL/6J mice with a low-iron diet (LID) for eight weeks before mating, while control mice were fed a normal diet (ND) (Figure 2A). The LID caused decreased growth in body weight (Figure 2B), a higher rate of embryo resorption (Figure 2C,D), and a reduction in placental and fetal weights (Figure 2E,F). Analysis of the DSCs from the pregnant mice revealed that the percent of calcein^+^ DSCs was lower, while the percent of ferritin^+^ DSCs was higher in the LID group (Figure 2G), suggesting LID-induced iron deposition in the decidua. In addition, *SLC40A1* expression was decreased, while *FTH* expression was increased in the uterus of the LID group (Figure 2H). Thus, a LID might induce a reduction in the iron discharge of DSCs, resulting in local iron deposition in the decidua and leading to adverse pregnancy outcomes.

### 2.3. Iron Supplementation Reduced Iron Deposition in Decidua of Abortion-Prone Mice and Improved Pregnancy Outcome

Both LID-fed mice and abortion-prone mice displayed local iron deposition in the decidua; thus, could iron supplementation ameliorate the pregnancy outcome of abortion-prone mice? The World Health Organization recommends routine iron supplementation of 30–60 mg per day throughout pregnancy [15], which, converted into the dose by intraperitoneal injection in mice, is about 0.025–0.05 mg per day. The recommended dose of iron dextran injection for the treatment of human iron-deficiency anemia is 100 mg every one to three days, which, converted into the dose by intraperitoneal injection of mice, is about 0.26 mg per one to three days. As low serum iron existed in RPL patients but they were non-anemic, the spontaneous-abortion-prone mice (female CBA/J mice mated with male DBA/2 mice) were injected intraperitoneally with PBS or 0.1 mg iron dextran (the upper limit of iron supplement dose recommended by WHO but less than that in the treatment of iron-deficiency anemia) every two days. There was no difference in the maternal weight (Figure 3A), fetal weight, or placental weight (Figure 3B) of normal pregnant mice (female CBA/J mice mated with male BALB/c mice), spontaneous-abortion-prone mice and spontaneous-abortion-prone mice supplemented with iron. Spontaneous-abortion-prone mice models displayed a high embryo resorption rate compared with normal pregnancy models. Iron supplementation reduced embryo absorption (Figure 3C,D) and relieved the local iron deposition in the decidua of spontaneous-abortion-prone mice (Figure 3E).

### 2.4. Iron Overload Caused Ferroptosis of DSCs

As iron deposition in DSCs seemed to be associated with adverse pregnancy outcome, we explored the effect of iron overload on DSCs. Both FeSO_4_ (ferrous iron) and ferric citrate (ferric iron) can enhance intracellular iron levels in DSCs, as calcein decreased after FeSO_4_ and ferric citrate treatment (Figure 4A and Appendix A). Cell damage was aggravated with an increase in FeSO_4_ and ferric citrate concentrations (Figure 4B and Appendix A). The mitochondrial function of DSCs was significantly impaired, as ATP production capacity decreased with an increase in FeSO_4_ concentration (Figure 4C). Mitochondrial function was further assessed by a Seahorse XF analyzer. As shown in Figure 4D and Appendix A, FeSO_4_ and ferric citrate inhibited the maximum respiratory capacity and respiratory potential of DSCs in a dose-dependent manner. Furthermore, FeSO_4_ and ferric citrate also increased the mitochondrial superoxide (Figure 4E and Appendix A), intracellular reactive oxygen species (ROS) (Figure 4F,G and Appendix A), and lipid ROS (Figure 4H and Appendix A) of DSCs.

The rising lipid ROS of DSCs attracted our attention, as lipid peroxidation was a functional marker for ferroptosis [16]. As expected, the normal morphology of the DSCs gradually decreased (Figure 5A and Appendix A) and cell death increased (Figure 5B and Appendix A) under FeSO_4_ or ferric citrate treatment, and these effects were reversed by ferrostatin-1, an inhibitor of ferroptosis (Figure 5C,D). Both *TRFC* overexpression (to induce cellular uptake of iron, Figure 5E) and *SLC40A1* downregulation (to inhibit iron export, Figure 5H) enhanced FeSO_4_-induced lipid ROS (Figure 5F,I) and cell death (Figure 5G,J) in DSCs. These data indicate that iron overload causes ferroptosis of DSCs in vitro.

### 2.5. Ferroptosis Occurred in the DSCs of Abortion

Next, we sought to determine whether ferroptosis was enhanced in human DSCs from pregnancies complicated by RPL when compared to normal early pregnancies. As shown in Figure 6A, the lipid ROS in DSCs were elevated in these RPL patients compared to normal controls. Cell death of DSCs was also augmented in RPL patients (Figure 6B). Similar results were observed with respect to the immune-response-mediated spontaneous abortion model (♀CBA/J × ♂DBA/2) and LPS-induced C57BL/6J abortion model (Figure 6C–F). Thus, ferroptosis was enhanced in the DSCs of pregnancies that resulted in abortion, in both the human and mouse model.

### 2.6. Cystine-GSH-GPX4 Axis Participated in the Regulation of Ferroptosis of DSCs

Glutathione peroxidase 4 (GPX4) is an antioxidative enzyme that is one of the central regulators of ferroptosis [17], and it is also expressed in the mammalian uterus [18]. Western blot analysis revealed that treatment with FeSO_4_ downregulated GPX4 protein levels in the DSCs (Figure 7A). *GPX4* overexpression (Figure 7B) inhibited FeSO_4_-induced lipid ROS (Figure 7C) and cell death (Figure 7D) in DSCs.

Given that suppression of glutathione (GSH) biosynthesis with subsequent inhibition or degradation of GPX4 activity both contribute to regulating the initiation and execution of ferroptosis [19], we measured intracellular GSH levels in DSCs exposed to FeSO_4_. As shown in Figure 7B, FeSO_4_ downregulated GSH levels in DSCs in a dose-dependent manner. Both GSH and cystine (a substrate for the synthesis of GSH) supplementation reduced FeSO_4_-induced lipid ROS and cell death in DSCs (Figure 7F–I). These data suggest that the cystine-GSH-GPX4 axis participates in the regulation of ferroptosis in DSCs.

### 2.7. Ferroptosis Inhibition Improved Pregnancy Outcomes in Two Abortion Models

To further determine whether ferroptosis modulation affected pregnancy outcomes, we invested the potential value of a ferroptosis inhibitor (liproxstatin-1) in preventing spontaneous abortion in vivo. Liproxstatin-1 effectively reversed embryo loss and reduced lipid ROS and cell death in DSCs from the spontaneous-abortion-prone mice (Figure 8A–D). Owing to the regulatory effects of the cystine-GSH-GPX4 axis on ferroptosis in DSCs, we further treated spontaneous-abortion-prone mice with cysteine or GSH. As shown in Figure 8A–D, remarkably, both cysteine and GSH also reversed embryo loss and reduced lipid ROS and cell death in DSCs. In parallel experiments, we explored the effects of liproxstatin-1, cysteine, and GSH on LPS-induced abortion. Consistent with the results in the spontaneous-abortion-prone mice, the relatively higher resorption, lipid ROS, and cell death in DSCs of the LPS-induced abortion model was substantially reduced after treatment with liproxstatin-1, cysteine, or GSH (Figure 8E–H). The results in Figure 8 demonstrate that ferroptosis inhibition has a therapeutic effect on embryo loss in both the spontaneous abortion model and LPS-induced abortion model.

## 3. Discussion

Despite the significant association between imbalances in iron metabolism and severe adverse pregnancy outcomes [9], little is known regarding the iron status of the decidua in the context of RPL. Here, we showed that the serum iron of RPL patients was much lower than that in normal early pregnancies, even though those RPL patients were non-anemic and maternal hemoglobin levels did not differ between the two groups. However, low serum iron and decidual local iron deposition existed simultaneously in RPL patients. This was further confirmed by mice fed with a LID, as the LID induced iron deposition in the decidua and caused an adverse pregnancy outcome in mice.

Several studies have revealed higher levels of circulating iron and local iron overload in preeclampsia [9,20]. It was striking, then, that low serum iron and decidual local iron deposition existed simultaneously in RPL patients. We further found that cellular iron-exporter Fpn expression in DSCs was decreased in both RPL patients and LID-fed mice, thus decreasing iron export from the DSCs. This might be the reason for local iron deposition in DSCs from low-serum-iron RPL patients and LID-fed mice. Recent studies have also demonstrated that in iron-deficient mice, placental Fpn expression significantly decreased during the whole gestational period, which compromised iron delivery to the fetus [21,22]. Maternal iron availability in the circulation is mediated by hepcidin, which interacts with Fpn [23]; however, whether Fpn expression in DSCs of RPL individuals is regulated by hepcidin requires further research. Additional studies are also required to determine the exact mechanism(s) of maternal serum iron and Fpn suppression during RPL. In addition, the peripheral ferritin and the soluble transferrin receptor content should be checked in the future to better explore whether low serum iron and decidual local iron deposition actually exist simultaneously in RPL patients.

Decidual local iron deposition also existed in the spontaneous abortion model and the LPS-induced abortion model. Iron supplementation reduced iron deposition in the decidua of the spontaneous abortion model and improved pregnancy outcome. Though it is common clinical practice to provide iron supplementation to pregnant women [15], it has also been shown that routine iron supplementation in iron-replete women does not translate into improved perinatal outcomes, but rather appears to be associated with significantly more adverse pregnancy events [20]. Iron supplementation beginning in mid-pregnancy had little to no effect on iron transfer and might even cause adverse effects, such as fetal growth restrictions [24]. However, most studies are based on healthy pregnancies, neglecting conditions with underlying placental or maternal abnormalities. Thus, more evidence-based medical research is needed to determine when and how much iron should be supplemented in pregnant women with a history of RPL, to prevent the occurrence of RPL. Anemia-cutoff should be regarded differently in normal versus pregnancies associated with iron deficiency (especially in RPL), where supplementation may be beneficial.

We found that iron overload can cause ferroptosis in DSCs, and ferroptosis was enhanced in the DSCs of pregnancies ending in abortion, both in the human and mouse models. Ferroptosis is a newly recognized mechanism of programmed cell death, characterized by iron accumulation and lipid-peroxidation-mediated cell membrane damage [16]. Investigating the role of ferroptosis in trophoblasts or placentas brings new insights into understanding the pathogenesis of preeclampsia and other placental-related diseases [12,13]. Uterine and placental ferroptosis also play a role in polycystic-ovary-syndrome-like pregnant rats with oxidative stress-related fetal loss [25]. Several genes are known to modulate the link between ferroptosis and the pathogenesis of miscarriage in humans [26]. DSCs affect the pregnancy microenvironment because of their role in the recruitment, differentiation, and function of immune cells as well as in tissue remodeling. Abnormal apoptosis, senescence, and arrested decidual growth can severely affect the biological role of the decidua, further resulting in tissue dysfunction in neighboring placental tissues [7,8,27]. In the present study, we provide evidence that enhanced ferroptosis in DSCs might also affect the pregnancy outcome and be associated with RPL. In addition, a ferroptosis inhibitor effectively reduced embryo loss in two mouse abortion models. At the molecular level, decreased GPX4 levels or decreased GPX4 activity induces ferroptosis through depletion of GSH and inhibition of lipid peroxidation in different tissues and cells [19]. In vivo knockout studies have shown that mice lacking the entire *GPX4* gene experience early embryonic lethality [28]. In accordance with a role of GPX4 in ferroptosis in vivo, we showed that iron-overload-induced DSC ferroptosis was associated with decreased GPX4 and GSH levels. In parallel to mitigating ferroptosis, eliminating circulating ROS, and improving fetal survival [12,13], we found that treatment with cysteine (a substrate for the biosynthesis of GSH) and GSH can attenuate DSC ferroptosis and effectively reverse embryo resorption in the spontaneous abortion model and LPS-induced abortion model. However, further studies are still required to determine the exact mechanism(s) of FeSO_4_ effects on cystine-GSH-GPX4 axis. Whether AcSL4-mediated lipid metabolism regulates ferroptosis [29] in DSCs also requires additional research.

Iron homeostasis is required for supporting maternal requirements, placental function, and fetal development, while dysregulated iron status is associated with the occurrences of several pathological conditions, as aberrant accumulation of intracellular iron leads to oxidative stress, which can subsequently promote or amplify ferroptosis [9], although iron supplementation reduced iron deposition in the decidua of the spontaneous abortion model and improved pregnancy outcome. The negative impact of improper iron supplementation on fetal survival in control pregnant rats has been previously reported [30]. Such information, together with a better understanding of low-serum-iron-induced DSC ferroptosis, is valuable in providing further clinical evaluation of the safety and efficacy of iron supplementation for women during pregnancy. Our findings also suggest that ferroptosis-inhibiting agents might broaden the therapeutic strategies for diseases that stem from lipotoxic tissue injury, such as decidua dysfunction that manifest as RPL. Whereas the clinical use of ferroptosis-mitigation strategies is still distant, future deployment of targeted ferroptosis therapeutics may serve to attenuate decidua dysfunction and its sequalae, including RPL.

In summary (Figure 9), we found that low serum iron and decidual local iron deposition existed simultaneously in RPL patients. Decreased ferroportin expression that inhibited the iron export from DSCs may be the reason that local iron deposition occurred in DSCs from low-serum-iron RPL patients and LID-fed mice. Iron supplementation reduced iron deposition in the decidua in the spontaneous abortion model and improved pregnancy outcome. Local iron overload caused ferroptosis in DSCs by downregulating GSH and GPX4 levels. Ferroptosis inhibitor, cysteine, or GSH supplementation effectively attenuated DSC ferroptosis and reversed embryo loss in the spontaneous abortion model and LPS-induced abortion model. Thus, decidual stromal cell ferroptosis associated with abnormal iron metabolism is implicated in the pathogenesis of recurrent pregnancy loss, and ferroptosis mitigation may be a potential therapeutic target for RPL patients.

## 4. Material and Methods

### 4.1. Human Samples

This study recruited subjects aged from 20 to 35 years old from the Obstetrics and Gynecology Hospital of Fudan University, China, between July 2020 and December 2022 (the clinical characteristics of enrolled subjects are summarized in Table 1). Whole peripheral blood and decidual tissues of human first-trimester pregnancies were obtained under fasting conditions from clinically normal pregnancies (terminated for non-medical reasons, had at least one successful pregnancy and no history of spontaneous abortions, n = 62) and miscarriages (diagnosed as RPL, excluding those resulting from endocrine, anatomic, genetic abnormalities, infection, etc., n = 38).

DSCs were obtained from decidual tissue digesting in DMEM/F-12 supplemented with collagenase type IV (1.0 mg/mL, CLS-1; Worthington Biomedical, Lakewood, NJ, USA) and DNase I (150 U/mL, Applichem, Darmstadt, Germany) as described previously [14].

### 4.2. Cell Treatment

Freshly isolated DSCs were cultured overnight in complete medium and further incubated in serum-free medium for 12 h, followed by stimulation with a range of concentrations of the FeSO_4_ or ferric citrate for 48 h. In some experiments, DSCs were treated with *SLC40A1*-specific siRNA (si-*SLC40A1*: 5′-CCGAUCAAGGUUCAUUCAATT-3′), or *TFRC* plasmid for 20 h using transfection reagent (L3000015, Invitrogen, Waltham, MA, USA) according to the manufacturer’s instructions.

### 4.3. Immunohistochemistry

Sections (5 mm) of paraffin-embedded first-trimester human decidua were incubated with rabbit anti-human TfR1 antibody (ab214039, Abcam, Waltham, MA, USA) or anti-human Fpn antibody (NBP1-21502, Novus, Centennial, CO, USA) overnight at 4 °C in a humidified chamber. After washing, sections were overlaid with anti-rabbit/mouse IgG HRP conjugate (Gene tech, Shnaghai, China) at room temperature for 1 h. The reaction was developed with 3,3-diaminobenzidine, and sections were counterstained with hematoxylin.

### 4.4. Western Blot [31]

The cell samples were lysed with cold radio-immunoprecipitation (RIPA) butter (Beyotime Biotechnology, Haimen, China) supplemented with a protein inhibitor cocktail (Roche, Branford, CT, USA, and a phosphatase inhibitor cocktail. Protein concentrations were determined through the BCA method. Lysates were heated at 95 °C for 5 min or at 60 °C for 30 min and then loaded on 10% gels (Bio-Rad, Hercules, CA, USA) for SDS–polyacrylamide gel electrophoresis. After electrophoretic separation, the proteins were transferred onto 0.2 μm PVDF membranes, blocked with 5% nonfat milk, and incubated overnight at 4 °C with the primary antibodies targeting anti-Fpn (NBP1-21502, Novus, Centennial, CO, USA), anti-TfR1 (13113, Cell Signaling Technology, Danvers, MA, USA), anti-GPX4 (ab125066, Abcam, Waltham, MA, USA), anti-β-actin (AA128, Biotechnology, Shanghai, China), β-actin used as internal standards. Membranes were washed and incubated with HRP-conjugated secondary antibody (Abmart, Shanghai, China) at room temperature for 1 h. The antibody-labeled proteins were detected by chemiluminescence using Chemiluminescent HRP Substrate in an Amersham™ Imager 600 (GE Healthcare, Chicago, IL, USA). β-actin was used to normalize the protein expression by optical intensity analysis using the ImageJ software V1.52 a (Wayne Rasband National Institutes of Health, Bethesda, ML, USA).

### 4.5. Quantitative Real-Time Polymerase Chain Reaction (qRT-PCR)

Total RNA was extracted from cells or homogenized tissues using TRIzol reagent (Invitrogen, Waltham, MA, USA) according to the manufacturer’s instructions. Complementary DNA (cDNA) was synthesized using PrimeScript™ RT Master Mix (Takara, Osaka, Japan) and then amplified using SYRB Green PCR Master Mix (Yeasen, Shanghai, China) with Applied Biosystems™ QuantStudio™ 6 (ThermoFisher Scientific, Waltham, MA, USA). *β-actin* was used as an internal control to normalize the relative changes in specific gene expression using the 2^−ΔΔCt^ method. Mouse primer sequences for QPCR were as follows: *TFRC*: forward 5′-GTTTCTGCCAGCCCCTTATTAT-3′ and reverse 5′-GCAAGGAAAGGATATGCAGCA-3′; *SLC40A1*: forward 5′-ATGGGAACTGTGGCCTTCAC-3′ and reverse 5′-TCCAGGCATGAATACGGAGA-3′; and *FTH:* forward 5′-CCATCAACCGCCAGATCAAC-3′ and reverse 5′-GAAACATCATCTCGGTCAAA-3′.

### 4.6. LDH Cytotoxicity Assay

The extracellular concentration of lactate dehydrogenase (LDH) was used to evaluate cell damage. LDH released by DSCs stimulated with FeSO_4_ and ferric citrate was determined by the CyQUANT™ LDH Cytotoxicity Assay Kit (ThermoFisher Scientific, Waltham, MA, USA) following the manufacturer’s instructions.

### 4.7. Analysis of ATP

The cellular level of ATP generated by DSCs stimulated by FeSO_4_ and ferric citrate was determined using an ATP Assay Kit (S0026, Beyotime Biotechnology, Shanghai, China), following the manufacturer’s protocol.

### 4.8. Oxygen Consumption Rate Analysis

The oxygen consumption rate (OCR) of DSCs was measured using a Seahorse XF96 analyzer [32]. For common tests, prepared cultured cells were washed with XF DMEM medium supplemented with 1 mM pyruvate, 2 mM glutamine, and 10 mM glucose, and preincubated at 37 °C for around 45 min in the absence of CO_2_. Proper concentration solutions of oligomycin, FCCP, and rotenone/antimycin A were loaded into the ports on the sensor cartridge, then the program was run according to the manufacturer’s protocol for the Seahorse XF Cell Mito Stress Test Kit (103015-100, Agilent, Beijing, China).

### 4.9. Intracellular ROS, Lipid ROS, and Mitochondrial Superoxide Measurement

The levels of intracellular ROS [33], lipid ROS [34], and Mitochondrial Superoxide [35] were indicated by 2,7-dichlorodihydro-fluorescein diacetate (DCFH-DA, D6883, Sigma, Burlington, MA, USA), C11 BODIPY™ 581/591 (D3861, ThermoFisher Scientific, Waltham, MA, USA), and MitoSOX™ Red Mitochondrial Superoxide Indicator (M36008, ThermoFisher Scientific, Waltham, MA, USA), respectively. After the indicated treatments, cells were stained with 5 μM DCFH-DA for 20 min, 5 μM C11 BODIPY™ 581/591 for 30 min, or 5 μM MitoSOX Red for 10 min at 37 °C in the dark and then washed twice with PBS. The fluorescence of cells in PBS was measured using a Beckman–Coulter CyAn ADP cytometer (Beckman-Coulter, Bria, CA, USA), and images were recorded using a fluorescence microscope (Olympus, Tokyo, Japan).

### 4.10. 7-Amino-Actinomycin D (7-AAD) Staining for Cell Death

Cell death was evaluated by 7-AAD staining [36] (BD Pharmingen, Franklin Lakes, IL, USA). DSCs were harvested and resuspended in 500 µL PBS containing 5 µM 7AAD working solution and then incubated in the dark for 30 min at 37 °C. The 7-AAD fluorescence of DSCs was determined immediately by flow cytometry (Beckman-Coulter, Bria, CA, USA) and analyzed with FlowJo software 7.6 (Tree Star, Ashland, OR, USA).

### 4.11. Analysis of GSH Content

The cellular level of glutathione was determined using a Reduced Glutathione Assay Kit [37] (A006-2-1, Jiancheng, Nanjing, China), following the manufacturer’s protocol.

### 4.12. Mice

CBA/J female, DBA/2 male, BALB/c male, C57BL/6J female, and C57BL/6J male mice were purchased from Shanghai SLAC Laboratory Animal Co., Ltd. (Shanghai, China) and Beijing HFK Bioscience Co., Ltd. (Beijing, China) and bred in a room at 22–25 °C, with 40–60% relative humidity and 14 h light–10 h dark cycles. Eight-week-old CBA/J females were mated to BALB/c males to provide normal pregnancy (NP) models. All females were inspected the next morning for vaginal plugs. The day of visualization of a plug was designated as day 0.5 of pregnancy (GD 0.5). Eight-week-old CBA/J females were mated to DBA/2 males to establish spontaneous-abortion-prone (SA) models [38]. For the LPS-induced abortion model (LPS), C57BL/6J females were mated with C57BL/6J males and intraperitoneally injected with 0.25 mg/kg LPS at GD 7.5 [38]. For the low-iron-diet model [39], female C57BL/6J mice were fed with a low-iron diet (0.9 ppm Fe, D08080402, Research Diets, New Brunswick, NJ, USA) for eight weeks before mating. In some groups, pregnant CBA/J mice of SA or pregnant C57BL/6J mice of LPS were administrated with 200 mg/kg GSH (Sigma, Burlington, MA, USA) or 10 mg/kg Liproxstain-1 (Sigma, Burlington, MA, USA) [40] by intraperitoneal injection, or 200 mg/kg cystine intragastrically on GD 2.5, 4.5, and 6.5. In some groups, pregnant CBA/J mice of SA were administrated with 0.1 mg iron dextran or PBS intraperitoneally every 2 days for two weeks. All pregnant mice were monitored at GD 14.5. The percentage of fetal loss (embryo absorption rate) was calculated as: % of resorption = R/(R + V) × 100, where R represents the number of hemorrhagic implantation (sites of fetal loss) and V stands for the number of viable, surviving fetuses.

### 4.13. Intracellular Iron Content Measurement

Intracellular iron concentration was evaluated using flow cytometry [41]. For calcein analysis, cells were incubated in DMEM/F12 medium containing 2 μM calcein-AM (Sigma, Burlington, MA, USA) for 15 min at 37 °C. For ferritin analysis, cells were first fixed and permeabilized by the Fix/Perm kit (Biolegend, San Diego, CA, USA) and then incubated with anti-ferritin antibody (Abcam ab75973, Waltham, MA, USA) for 45 min at 4 °C in the dark. After being washed twice, cells were incubated for 45 min with Alexa Flour 594-conjugated anti-Rabbit IgG (Abcam, Waltham, MA, USA). APC/CY7-conjugated anti-mouse CD45 antibodies (Biolegend, San Diego, CA, USA) and Alexa Fluor 647-conjugated anti-mouse Vimentin (Biolegend, San Diego, CA, USA) were also used. Flow cytometry was performed on a Beckman-Coulter CyAn ADP cytometer (Beckman-Coulter, Bria, CA, USA) and analyzed with FlowJo software 7.6 (Tree Star, Ashland, OR, USA).

### 4.14. Statistical Analysis

All variables were normally distributed in this study. Thus, variables are presented as means and standard error of the mean (SEM). One-way analysis of variance (ANOVA) was used to evaluate differences. A *p*-value of less than 0.05 was considered statistically significant. For variables with a *p*-value of less than 0.05 in ANOVA, a post hoc Dunnett *t*-test was performed to determine differences between each group. All analyses were carried out using GraphPad Prism 8 software (GraphPad, San Diego, CA, USA).

## Figures and Tables

**Figure 1 ijms-24-07836-f001:**
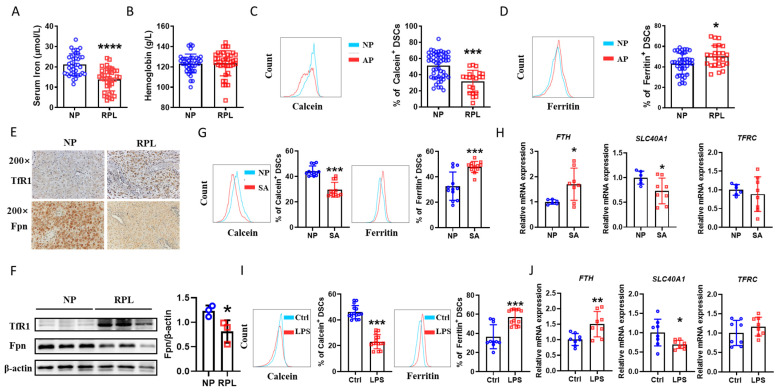
Iron deposition in DSCs of abortion. (**A**) Serum iron of normal pregnancy (NP, n = 38) and patients who were diagnosed with recurrent pregnancy loss (RPL, n = 38). (**B**) Peripheral blood hemoglobin of NP (n = 38) and RPL (n = 38). (**C**) Frequency of calcein-expressing DSCs from NP (n = 51) and RPL (n = 21). (**D**) Frequency of ferritin-expressing DSCs from NP (n = 39) and RPL (n = 24). (**E**) Immunohistochemical localization of transferrin receptor 1 (TfR1) and ferroportin (Fpn) in decidual tissue from NP and RPL. (**F**) Protein level of TfR1 and Fpn in DSCs from NP and RPL. (**G**) Frequency of calcein- and ferritin-expressing DSCs from normal pregnant (NP, n = 12) and spontaneous-abortion-prone (SA, n = 12) mice. (**H**) Real-time PCR analysis of *FTH*, *SLC40A1*, and *TFRC* expression in the uterus of NP and SA mice. (**I**) Frequency of calcein- and ferritin-expressing DSCs from pregnant C57BL/6J mice (Ctrl, n = 15) and LPS-treated pregnant C57BL/6J mice (LPS, n = 15). (**J**) Real-time PCR analysis of *FTH*, *SLC40A1,* and *TFRC* expression in the uterus of pregnant C57BL/6J mice and LPS-treated pregnant C57BL/6J mice. Images are representative of three individual experiments. Data represent the mean ± standard error of the mean (SEM) and are representative of three independent analyses. * *p* < 0.05, ** *p* < 0.01, *** *p* < 0.001, **** *p* < 0.0001.

**Figure 2 ijms-24-07836-f002:**
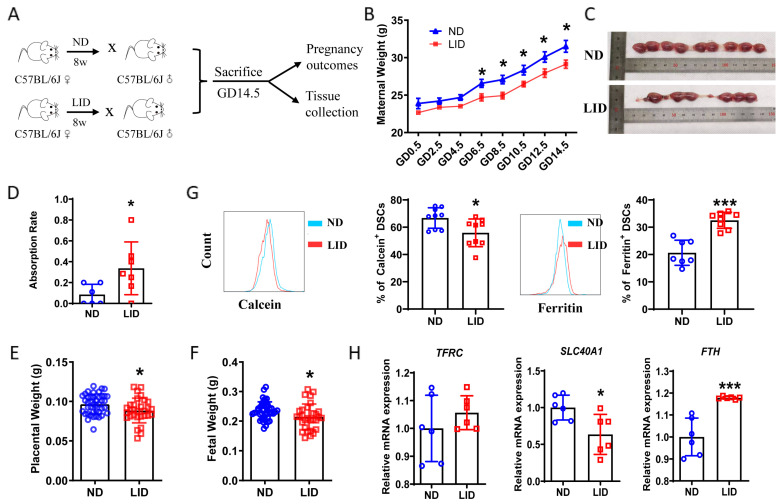
Low-iron-diet-induced iron deposition in the decidua and adverse pregnancy outcome in mice. (**A**) Schematic diagram showing the experimental design of the mouse models involving the establishment, gestational termination at GD 14.5, pregnancy outcome observation, tissue collection for cell isolation, and analysis. (**B**) Weight of pregnant C57BL/6J females fed with a normal diet (ND) or low-iron diet (LID). (**C**) Representative images of the uterus from pregnant C57BL/6J females fed with ND or LID. (**D**–**F**) Percent of fetal resorption (**D**), placental weights (**E**), and fetal weights (**F**) of pregnant C57BL/6J females fed with ND or LID. (**G**) Frequency of calcein- and ferritin-expressing DSCs from pregnant C57BL/6J mice fed with ND or LID. (**H**) Real-time PCR analysis of *FTH*, *SLC40A1,* and *TFRC* expression in the uterus of pregnant C57BL/6J mice fed with ND or LID. Data represent the mean ± SEM of n = 3–9 mice per group and are representative of three independent analyses. * *p* < 0.05, *** *p* < 0.001.

**Figure 3 ijms-24-07836-f003:**
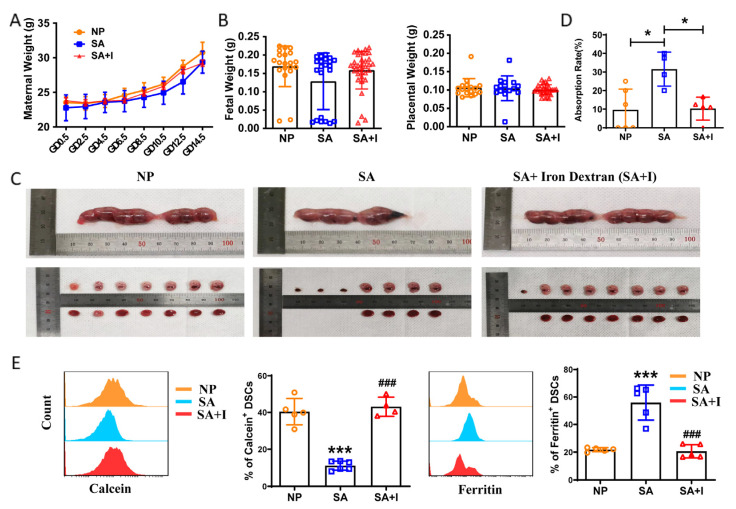
Iron supplementation reduced iron deposition in the decidua of abortion-prone mice and improved pregnancy outcome. (**A**) Weight of pregnant females from normal pregnant (NP) mice, spontaneous-abortion-prone (SA) mice, and SA mice supplemented with iron (SA + I). (**B**) Fetal and placental weights of pregnant female NP, SA, and SA + I mice. (**C**) Representative images of the uterus from pregnant female NP, SA, and SA + I mice. (**D**) Percent of fetal resorption of pregnant female NP, SA, and SA + I mice, * *p* < 0.05. (**E**) Frequency of calcein- and ferritin-expressing DSCs from pregnant female NP, SA, and SA + I mice. Data represent the mean ± SEM of n = 3–8 mice per group and are representative of three independent analyses. *** *p* < 0.001, compared with the NP group. ### *p* < 0.001, compared with the SA group.

**Figure 4 ijms-24-07836-f004:**
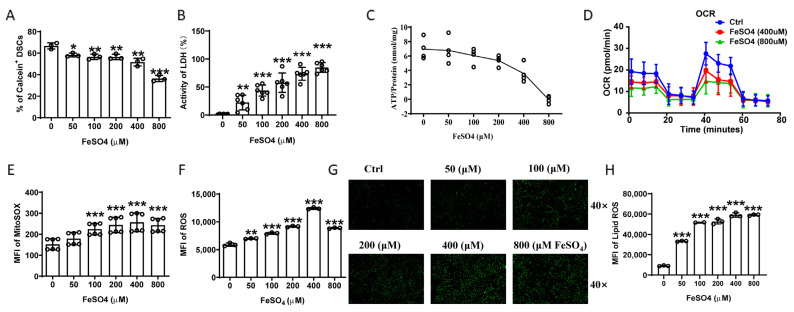
Iron overload caused DSC damages. (**A**) Quantification of flow cytometric analysis of calcein expression in DSCs stimulated with the indicated concentrations of FeSO_4_. (**B**) Extracellular concentration of LDH, used to evaluate cell damage. (**C**) ATP generation ratio detected by a luminometer. (**D**) Oxygen consumption rate (OCR) of DSCs stimulated with the indicated concentrations of FeSO_4_. (**E**) Levels of DSC mitochondrial superoxide detected by flow cytometry using a MitoSOX Red mitochondrial superoxide indicator. Summary of mean fluorescent intensity (MFI) from three independent experiments. (**F**) Intracellular ROS detected by fluorescent probe DCFH-DA in DSCs stimulated with the indicated concentrations of FeSO_4_. (**G**) Representative images of DSC intracellular ROS detected by fluorescence microscopy. Images are representative of three individual experiments. (**H**) The levels of intracellular lipid ROS of DSCs were detected by flow cytometry using a C11 BODIPY™ 581/591 lipid peroxidation sensor. Data represent the mean ± SEM and are representative of three independent analyses. * *p* < 0.05, ** *p* < 0.01, *** *p* < 0.001.

**Figure 5 ijms-24-07836-f005:**
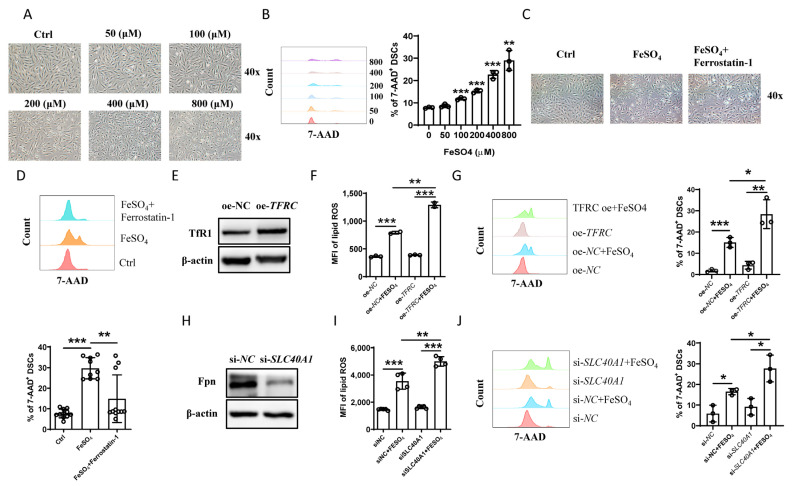
Iron overload caused ferroptosis of DSCs. (**A**) Representative images of DSC death stimulated with the indicated concentrations of FeSO_4_ observed by Inverted Phase Contrast Microscope (IPCM). (**B**) Cell death was evaluated based on 7-AAD expression in DSCs stimulated with the indicated concentrations of FeSO_4_ by flow cytometry. (**C**) Representative images of DSC death observed by IPCM. (**D**) Flow cytometric analysis (**upper**) and quantitation (**lower**) of death based on 7-AAD expression in DSCs stimulated with FeSO_4_ (400 μM) in the presence or absence of ferrostatin-1. (**E**) Protein level of TfR1 in DSCs with or without *TfR1* overexpression. (**F**) Intracellular lipid ROS of DSCs stimulated by FeSO_4_ (400 μM) with or without *TfR1* overexpression. Summary of MFI from three independent experiments. (**G**) Flow cytometric analysis (**left**) and quantitation (**right**) of death based on 7-AAD expression in DSCs stimulated by FeSO_4_ (400 μM) with or without *TfR1* overexpression. (**H**) Protein level of Fpn in DSCs with or without *SLC40A1* knockdown. (**I**) Intracellular lipid ROS of DSCs stimulated by FeSO_4_ (400 μM) with or without *SLC40A1* knockdown. (**J**) Flow cytometric analysis (**left**) and quantitation (**right**) of 7-AAD expression in DSCs stimulated by FeSO_4_ (400 μM) with or without *SLC40A1* knockdown. Data represent the mean ± standard error of the mean (SEM) and are representative of three independent analyses. * *p* < 0.05, ** *p* < 0.01, *** *p* < 0.001.

**Figure 6 ijms-24-07836-f006:**
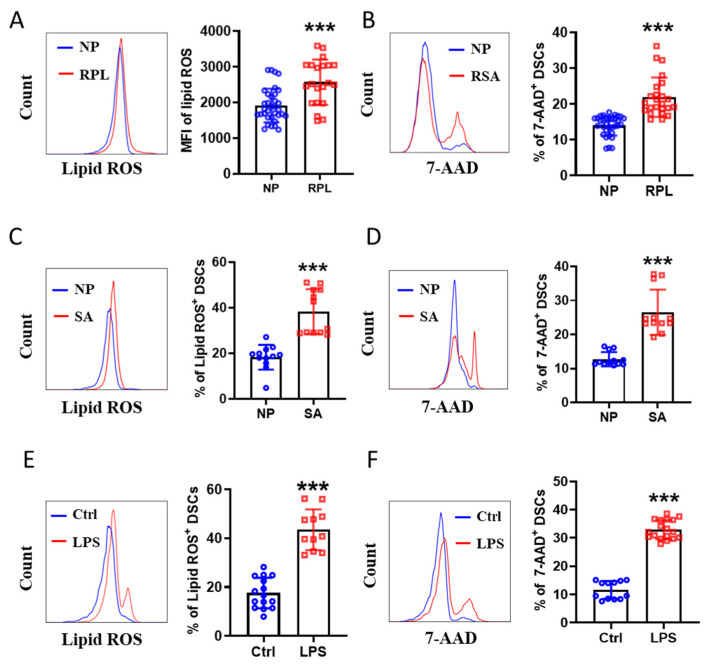
Ferroptosis occurred in the DSCs of abortion. (**A**) Levels of intracellular lipid ROS in DSCs from normal pregnancy (NP, n = 34) and patients who were diagnosed with recurrent pregnancy loss (RPL, n = 21). (**B**) Flow cytometric analysis (**left**) and quantitation (**right**) of 7-AAD expression in DSCs from NP (n = 33) and RPL (n = 24). (**C**,**D**) Levels of intracellular lipid ROS (**C**) and 7-AAD expression (**D**) of DSCs from normal pregnant (NP, n = 12) and spontaneous-abortion-prone (SA, n = 12) mice. (**E**,**F**) Levels of intracellular lipid ROS (**E**) and 7-AAD expression (**F**) of DSCs from pregnant C57BL/6J mice (Ctrl, n = 12–15) and LPS-treated pregnant C57BL/6J mice (LPS, n = 11–18). Data represent the mean ± SEM and are representative of three independent analyses, *** *p* < 0.001.

**Figure 7 ijms-24-07836-f007:**
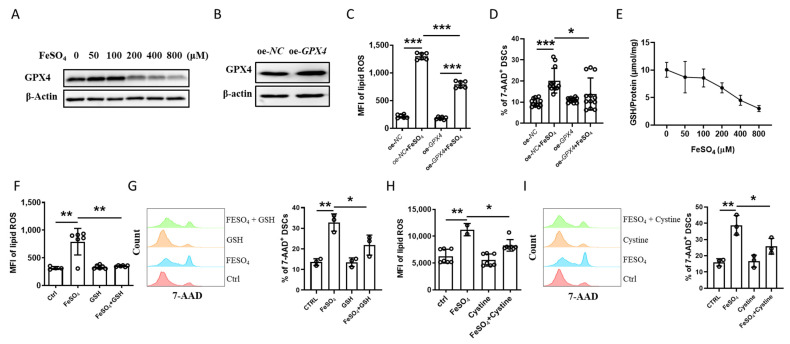
Cystine-GSH-GPX4 axis participated in the regulation of ferroptosis of DSCs. (**A**) Protein levels of GPX4 in DSCs stimulated with the indicated concentrations of FeSO_4_. (**B**) Protein levels of GPX4 in DSCs with or without *GPX4* overexpression. (**C**,**D**) Levels of intracellular lipid ROS (**C**) and 7-AAD expression (**D**) in DSCs stimulated by FeSO_4_ (400 μM) with or without *GPX4* overexpression. (**E**) GSH content in DSCs stimulated with the indicated concentrations of FeSO_4_. (**F**,**G**) Levels of intracellular lipid ROS (**F**) and 7-AAD expression (**G**) in DSCs stimulated by FeSO_4_ (400 μM) in the presence or absence of GSH. (**H**,**I**) Levels of intracellular lipid ROS (**H**) and 7-AAD expression (**I**) in DSCs stimulated by FeSO_4_ (400 μM) in the presence or absence of cystine. Data represent the mean ± SEM and are representative of three independent analyses, * *p* < 0.05, ** *p* < 0.01, *** *p* < 0.001.

**Figure 8 ijms-24-07836-f008:**
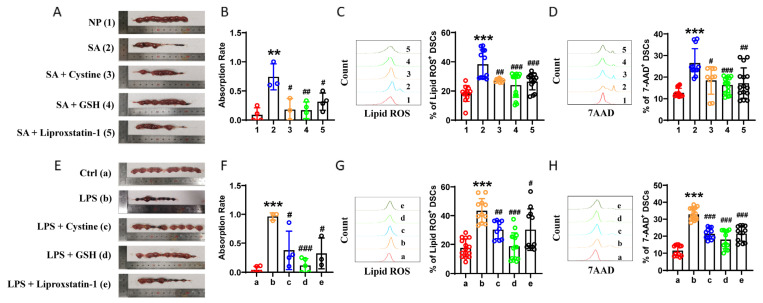
Ferroptosis inhibition improved pregnancy outcomes in two abortion models. (**A**,**B**) Representative images of the uterus (**A**) and percent of fetal resorption (**B**) of pregnant females of normal pregnant (NP) mice, spontaneous-abortion-prone (SA) mice, SA mice supplemented with cystine (SA + Cystine), SA mice supplemented with GSH (SA + GSH), and SA mice supplemented with liproxstatin-1 (SA + Liproxstatin-1). (**C**,**D**) Levels of intracellular lipid ROS (**C**) and 7-AAD expression (**D**) in DSCs from NP, SA, SA + Cystine, SA + GSH, and SA + Liproxstatin-1 mice. Data represent the mean ± SEM of n = 3–15 mice per group and are representative of three independent analyses. ** *p* < 0.01, *** *p* < 0.001, compared with the NP group. # *p* < 0.05, ## *p* < 0.01, ### *p* < 0.001, compared with the group SA. (**E**,**F**) Representative images of uterus (**E**) and percent of fetal resorption (**F**) of pregnant females of C57BL/6J mice (Ctrl, n = 15) and LPS-treated pregnant C57BL/6J mice (LPS, n = 15), LPS mice supplemented with cystine (LPS + Cystine), LPS mice supplemented with GSH (LPS + GSH), and LPS mice supplemented with liproxstatin-1 (LPS + Liproxstatin-1). (**G**,**H**) Levels of intracellular lipid ROS (**G**) and 7-AAD expression (**H**) of DSCs from Ctrl, LPS, LPS + Cystine, LPS + GSH, and LPS + Liproxstatin-1 mice. Data represent the mean ± SEM of n = 3–8 mice per group and are representative of three independent analyses. *** *p* < 0.001, compared with the control group. # *p* < 0.05, ## *p* < 0.01, ### *p* < 0.001, compared with the group LPS.

**Figure 9 ijms-24-07836-f009:**
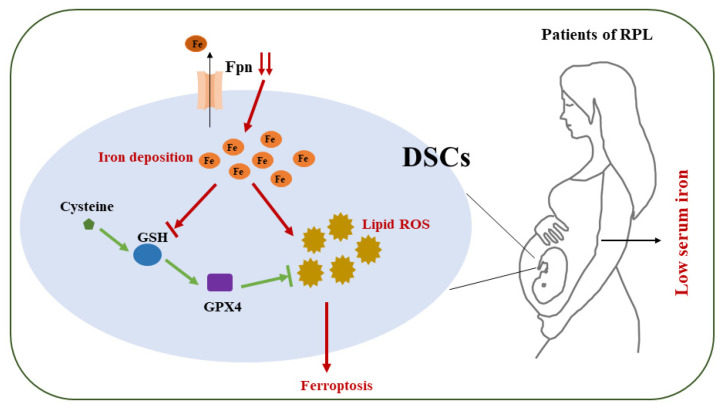
Schematic diagram showing decidual stromal cell ferroptosis associated with abnormal iron metabolism implicated in RPL. Low serum iron and decidual local iron deposition exist simultaneously in RPL patients. Decreased Fpn expression that inhibits iron export from DSCs might be the reason that local iron deposition occurs in DSCs from low-serum-iron RPL patients and low-iron-diet-fed mice. Local iron overload caused ferroptosis in DSCs by downregulating GSH and GPX4 levels. Thus, ferroptosis in DSCs associated with abnormal iron metabolism might be implicated in the pathogenesis of RPL, and ferroptosis mitigation might be a potential therapeutic target for RPL patients.

**Table 1 ijms-24-07836-t001:** Clinical characteristics of enrolled subjects.

Subjects	NP	RPL	P
Number	62	38	ns
Age mean(years) ^a^	30.10 ± 0.48	30.00 ± 0.59	ns
Age range(years)	20–35	22–35	ns
Previous spontaneous abortion (number) ^a^	-	2.82 ± 0.16	ns
Pregnancy day (venous blood was collected) ^a^	43.29 ± 0.51	42.97 ± 0.54	ns
Treatment history	-	-	-
Serum Iron (μmol/L)	21.16 ± 0.86	13.76 ± 0.91	<0.0001
Hemoglobin (g/L)	123.3 ± 1.51	123.5 ± 1.99	ns

Normal pregnancy (NP), human first-trimester pregnancies were obtained from clinically normal pregnancies (terminated for non-medical reasons, had at least one successful pregnancy and no history of spontaneous abortions); Recurrent pregnancy loss (RPL), patients diagnosed as RPL, excluding those resulting from endocrine, anatomic, genetic abnormalities, infection, etc. ^a^ Median ± standard error of the mean (SEM). No significance (ns).

## Data Availability

All data presented in this study are included in this published article and its Appendix A.

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
