# Peer review of "Decidual Stromal Cell Ferroptosis Associated with Abnormal Iron Metabolism Is Implicated in the Pathogenesis of Recurrent Pregnancy Loss"

_ijms, 2023, doi:10.3390/ijms24097836_

Round 1
Reviewer 2 Report
Methodological Biases exist
(The Authors must see my remarks)

Round 2
Reviewer 1 Report
I have the following comments:
1. Was the patient’s blood taken under fasting or fed conditions? Not resolved
2. Did you measure ferritin and the soluble transferrin receptor in the patient blood? Mentioned as a limitation, accepted to be resolved
3. The optical density (OD) of the Fpn and beta-actin bands in Figure 1 seems to be correlated, which is especially the case for sample number 6. Did you quantify the OD and correct Fpn OD with beta-actin OD? Not resolved.
4. Regulation of Fpn is predominantly post-transcriptional. Do you have Fpn data on protein level corresponding to RNA data in figure 2, for ND and LID mice? Not resolved
5. You write twice in the manuscript and in the legend of figure 8 about embryo absorption. Is this a spelling error? I am not familiar with this model, accepted to be resolved.
6. You address activation of the GPX4 axis. Did you also measure SLC7A11 (Cystin-Glutamin-Carrier) and AcSL4: Long-chain-fatty-acid—CoA ligase 4 in this context? Mentioned as a limitation, accepted to be resolved

Author Response
- Was the patient’s blood taken under fasting or fed conditions? Not resolved
Response:Thank you so much for your kind comments. The patients’s blood were taken under fasting conditions. We added this information in the revised manuscript. Line 408-409.
- Did you measure ferritin and the soluble transferrin receptor in the patient blood? Mentioned as a limitation, accepted to be resolved
Response:Thank you so much for your kind comments.
- The optical density (OD) of the Fpn and beta-actin bands in Figure 1 seems to be correlated, which is especially the case for sample number 6. Did you quantify the OD and correct Fpn OD with beta-actin OD? Not resolved.
Response:Thank you so much for your helpful comments. We quantified the OD and corrected Fpn OD with β-actin OD. The results showed that Fpn expression in RPL was decreased on protein level compared that in NP. We added this result in the revised manuscript. Figure 1F, line 450-452.
- Regulation of Fpn is predominantly post-transcriptional. Do you have Fpn data on protein level corresponding to RNA data in figure 2, for ND and LID mice? Not resolved
Response:Thank you so much for your helpful comments. The western blot analysis revealed that Fpn expression was decreased in the uterus of LID group (Figure in the below). However, these data were from another project, sorry that we could not put related data in this manuscript. We discussed this in the revised manuscript. Line 134-135.
- You write twice in the manuscript and in the legend of figure 8 about embryo absorption. Is this a spelling error?I am not familiar with this model, accepted to be resolved.
Response:Thank you so much for your kind comments.
- You address activation of the GPX4 axis. Did you also measure SLC7A11 (Cystin-Glutamin-Carrier) and AcSL4: Long-chain-fatty-acid—CoA ligase 4 in this context? Mentioned as a limitation, accepted to be resolved
Response:Thank you so much for your kind comments.
